

# Updated merged SAGE - CCI- OMPS+ dataset for evaluation of ozone trends in the stratosphere

Viktoria F. Sofieva[1], Monika Szelag[1], Johanna Tamminen[1], Carlo Arosio[2], Alexei Rozanov[2], Mark Weber[2], Doug Degenstein[3], Adam Bourassa[3], Daniel Zawada[3], Michael Kiefer[4], Alexandra Laeng[4], Kaley A. Walker[5], Patrick Sheese[5], Daan Hubert[6], Michel van Roozendael[6], Christian Retscher[7], Robert Damadeo[8], Jerry D. Lumpe[9]

[1] Finnish Meteorological Institute, Helsinki, Finland
[2] Institute for Environmental Physics, University of Bremen, Bremen, Germany
[3] Institute of Space and Atmospheric Studies, University of Saskatchewan, Saskatoon, Canada
[4] Karlsruhe Institute of Technology, Institute of Meteorology and Climate Research, Karlsruhe, Germany
[5] Department of Physics, University of Toronto, Toronto, Canada
[6] Royal Belgian Institute for Space Aeronomy (BIRA-IASB), Brussels, Belgium
[7] ESA/ESRIN, Frascati, Italy
[8] NASA Langley Research Center, Hampton, VA, USA
[9] Computational Physics Inc., Boulder, CO, USA

*Correspondence to*: Viktoria Sofieva (viktoria.sofieva@fmi.fi)

**Abstract.**

In this paper, we present the updated SAGE-CCI-OMPS+ climate data record of monthly zonal mean ozone profiles. This dataset covers the stratosphere and combines measurements by 9 limb and occultation satellite instruments – SAGE II, OSIRIS, MIPAS, SCIAMACHY, GOMOS, ACE-FTS, OMPS-LP, POAM III and SAGE III/ISS. Compared to the original version of the SAGE-CCI-OMPS dataset (Sofieva et al., 2017), the update includes new versions of MIPAS, ACE-FTS, and OSIRIS datasets, and introduces data from additional sensors (POAM III, SAGE III/ISS) and retrieval processors (OMPS-LP).

In the paper, we show detailed intercomparisons of ozone profiles from different instruments and data versions, with a focus on the detection of possible drifts in the datasets. The SAGE-CCI-OMPS+ dataset has a better coverage of polar regions and of the upper troposphere and the lower stratosphere (UTLS) than the SAGE-CCI-OMPS dataset.

We also studied the influence of including new datasets on ozone trends, which are estimated using multiple linear regression. The changes in the merged dataset do not change the overall morphology of post-1997 ozone trends: statistically significant trends are observed in the upper stratosphere. The largest changes in ozone trends are observed in polar regions, especially in the Southern Hemisphere.

The updated SAGE-CCI-OMPS+ dataset contains profiles of deseasonalized anomalies and ozone concentrations from 1984 to 2021, in 10° latitude bins from 90°S to 90°N, and in the altitude range from 10 km to 50 km. The dataset is in open access at https://climate.esa.int/en/projects/ozone/data/ and at ftp://cci_web@ftp-ae.oma.be/esacci (ESA Climate Office, last access: 10 August 2022).



## 1    Introduction

The importance of monitoring the stratospheric ozone and its vertical structure is nowadays well recognized, because ozone protects life on Earth from harmful ultraviolet solar radiation. Ozone evolution is connected with climate change, since ozone affects the radiation budget of the atmosphere (Brasseur and Solomon, 2005; WMO, 2018).  Recent studies  of ozone

trends (Weber et al., 2022; WMO, 2018 and references therein) have reported positive ozone trends in the upper stratosphere, as an expected consequence of international agreements on the reduction of ozone-depleting substances by the Montreal Protocol and its Amendments.

Satellite data play an important role in studies of ozone recovery and related processes. The main advantages of the satellite data are their global coverage and good accuracy. Since the temporal coverage of ozone data from individual satellite

instruments is limited, the data from several instruments are usually merged into long-term ozone climate data records.  Several merged datasets of ozone profiles and total columns have been used in recent analyses of ozone trends (e.g., Godin-Beekmann et al., 2022; Weber et al., 2022; WMO, 2018; Steinbrecht et al., 2017; Petropavlovskikh et al., 2019  and references therein).

One of the merged dataset of ozone profiles, which was used in the abovementioned studies, is the SAGE-CCI-OMPS dataset (Sofieva et al., 2017), which was created in the framework of the European Space Agency (ESA) Ozone Climate

Change Initiative (Ozone_cci, https://climate.esa.int/en/projects/ozone/). The SAGE-CCI-OMPS dataset  was derived from data by seven limb and occultation satellite instruments: MIPAS (Michelson Interferometer for Passive Atmospheric Sounding), SCIAMACHY(SCanning Imaging Spectrometer for Atmospheric CHartographY) and GOMOS(Global Ozone Monitoring by Occultation of Stars) on Envisat, OSIRIS(Optical Spectrograph and InfraRed Imaging System) on Odin, ACE-FTS (Atmospheric Chemistry Experiment Fourier Transform Spectrometer) on SCISAT,  OMPS-LP(Ozone Monitor Profiling

Suite-Limb Profiler) on Suomi-NPP, and SAGE II (Stratospheric Aerosol and Gases Experiment II) on ERBS.  The monthly zonal mean ozone profile dataset is provided in the altitude range from 10 to 50 km in 10° latitude bins. The merging is performed using deseasonalized anomalies. The original period of the SAGE-CCI-OMPS merged time series of ozone profiles was from late 1984 until the end of 2016, but it is regularly extended, and the latest version is available until the end of 2020. It is referred to as SAGE-CCI-OMPS throughout the paper.

This paper is dedicated to the updated version of the SAGE-CCI-OMPS dataset (referred to as SAGE-CCI-OMPS+ hereafter) and to related data intercomparisons.  The SAGE-CCI-OMPS+ dataset includes new versions of MIPAS, ACE-FTS and OSIRIS datasets and new data from POAM III (Polar Ozone and Aerosol Measurement) on SPOT 4, OMPS-LP processed by University of Bremen and SAGE III on the International Space Station (ISS).

The paper is organized as follows. Section 2 describes the ozone datasets from the individual instruments with the focus

on new versions or new datasets included in the SAGE-CCI-OMPS+ dataset.  Section 3 is dedicated to data merging for the SAGE-CCI-OMPS+ dataset and its updated version as well as to intercomparisons of the datasets. In Section 4, we analyze the sensitivity of trend analysis to the inclusion of new datasets. The conclusions are summarized in Section 5.


## 2 Data

### 2.1 Overview of the ozone datasets

For creating the merged SAGE-CCI-OMPS+ dataset, we use the data from several limb and occultation instruments, for which ozone profiles are retrieved on the geometric altitude grid. In the resulting merged dataset, ozone profiles are also

5 presented on the altitude grid from 10 to 50 km. The ozone profiles from individual instruments have a vertical resolution of 1–3 km in the stratosphere and in the UTLS (upper troposphere and the lower stratosphere). The information about individual datasets is collected in Table 1. The majority of the datasets – SAGE II, SAGE III, POAM III, GOMOS, OSIRIS, SCIAMACHY and OMPS – provide number density ozone profiles; therefore this representation is adopted for the merged dataset. For ACE-FTS and MIPAS, the retrievals are in volume mixing ratio on an altitude grid. Conversion to number density

10 profiles is performed using temperature profiles retrieved by these instruments as was done for the SAGE-CCI-OMPS dataset. For all instruments, we used ozone profiles datasets from the user-friendly HARMonized dataset of OZone profiles (HARMOZ) (Sofieva et al., 2013) developed in the Ozone_cci project. HARMOZ consists of the original retrieved ozone profiles from each instrument, which are screened for invalid data and presented on a common vertical grid and in a common netCDF4 format. In this work, we used altitude gridded datasets (HARMOZ_ALT), they are available at

15 https://climate.esa.int/en/projects/ozone/data/ and ftp://cci_web@ftp-ae.oma.be/esacci.

Four datasets (GOMOS, SCIAMACHY, SAGE II, and OMPS-LP processed by University of Saskatchewan) are the same as those used in SAGE-CCI-OMPS dataset (Sofieva et al., 2017). The detailed information about these datasets can be found in (Sofieva et al., 2013, 2017).Below we present the information about the new versions of the datasets and new datasets used in the SAGE-CCI-OMPS+ dataset.

**Table 1. Information about the datasets used in the SAGE-CCI-OMPS + dataset. Green color indicate new versions of the dataset, blue color indicate new datasets used (compared to the SAGE-CCI-OMPS dataset)**

| Instrument/ satellite | Processor, data source | Time period | Local time | Vertical resolution | Estimated precision | Profiles per day |
|---|---|---|---|---|---|---|
| SAGE II/ ERBS | NASA v7.0, HARMOZ_ALT | Oct 1984 – Aug 2005 | sunrise, sunset | ~1 km | 0.5–5% | 14–30 |
| OSIRIS/ Odin | USask v7.2, HARMOZ_ALT | Nov 2011 – present | 6 a.m., 6 p.m. | 2–3 km | 2-10% | ~250 |
| GOMOS/ Envisat | ALGOM2s v1.0, HARMOZ_ALT | Aug 2002 – Aug 2011 | 10 p.m. | 2–3 km | 0.5–5 % | ~110 |
| MIPAS/ Envisat | KIT/IAA V8, HARMOZ_ALT | Jan 2005 – Apr 2012 | 10 p.m., 10 a.m. | 3–5 km | 1–4% | ~1000 |
| SCIAMACHY/ Envisat | UBr v3.5, HARMOZ_ALT | Aug 2003- Apr 2012 | 10 a.m. | 3–4 km | 1–7% | ~1300 |
| ACE-FTS/ SCISAT | V4.1/4.2, HARMOZ_ALT | Feb 2004 – present | sunrise, sunset | ~3 km | 1–4% | ~30 |
| OMPS-LP/ Suomi NPP | USask 2D v1.1.0 UBr v3.3 HARMOZ_ALT | Apr 2012–present | 1:30 p.m. | ~1 km | 2–10% | ~1600 |



| | . | | | | | |
|---|---|---|---|---|---|---|
| SAGE III /ISS | NASA, AO3 v5.2 HARMOZ_ALT | 2017 – present | sunrise, sunset | ~1 km | 2–4% | ~30 |
| POAM III /SPOT 4 | NASA v.4 HARMOZ_ALT | 1998–2005 | sunrise, sunset | ~ 1 km stratosphere, 2-3 km upper troposphere | 3–5 % | ~30 |

### 2.2 ACE-FTS v4

The ACE-FTS instrument (Bernath et al., 2005) flies on board the Canadian SCISAT satellite, which was launched in 2003 into a non-sun synchronous, high-inclination orbit. The spectrometer is characterized by a high-spectral-resolution (0.02 cm$^{-1}$) and views the Earth's limb in the infrared spectral range between 750 and 4400 cm$^{-1}$. From its measurements it has been

possible to derive volume mixing ratio (VMR) profiles of over 40 atmospheric trace gases, more than 20 isotopologue species, together with pressure and temperature information. ACE-FTS observes the atmospheric limb between 5 and 150 km with a vertical sampling of ~ 2 to 6 km, depending on the orbital geometry and tangent height.

Recently an improved retrieval version of the dataset has been released and the ozone profiles have been validated against several independent observations (Sheese et al., 2022). The most recent retrieval algorithm is described in (Boone et al., 2020),

where the authors compare it to the previous v3.6 product. Version 4.1 has seen an update in spectroscopic information, including a new instrumental line shape, which improved the accuracy of forward model calculations. In addition, a 100 m sub-grid was introduced within each 1-km layer of the vertical grid. To reduce systematic errors found in previous processing versions, changes were introduced in the handling of solar and deep space calibration spectra.

Sheese et al. (2022) showed that v4.1 ozone data bias with respect to data sets is more stable with time in comparison to the

previous version, i.e., the drift affecting the v3.6 data is substantially reduced in v4.1. In the lower stratosphere, v4.1 data have a bias on the order of -1 % to +5 %, with a drift within ±1 % per decade. In the middle stratosphere, a positive bias of 2 % to 9 % was found, although the time series has very good stability, with a drift within ±0.5 % per decade. Finally, in the upper stratosphere, v4.1 ozone shows a positive bias that increases with altitude (up to ~15 %) with a drift within ±1 % per decade. Estimates indicate that the current product has a precision on the order of 0.1–0.2 ppmv below 20 km and above 45 km (~5–

10 %, depending on altitude). Between 20 and 45 km, the estimated random uncertainty is ~1–4 %.

### 2.3 MIPAS KIT v8

In the MIPAS IMK/IAA v8 data processing, the v8 Level1b dataset with improved characterization of detector ageing (improved detector non-linearity correction, see Sect. 5.6 of (Kleinert et al., 2018)) is used. This leads to less instrumentally caused drift in the retrieved ozone values. In addition, the v8 temperature retrieval is improved due to use of better a priori

information (Kiefer et al., 2021). The temperature retrieval results are used in the ozone retrieval. These improvements are relevant especially for the upper stratosphere and for the mesosphere. In addition, ozone retrievals include the 3D-structure of temperature retrieved in a previous step, thus errors due to horizontal inhomogeneities are reduced (Kiefer et al., 2022).



Comparisons with ACE-FTS, MLS and ozonesonde show approximately the same quality of V8 ozone as for V7. It is expected that the long-term stability of V8 is better than V7, particularly in the upper stratosphere (Laeng et al., Validation of final ozone product from MIPAS KIT/IAA scientific processor, 2022, in preparation).

## 2.4    OSIRIS v7.2

OSIRIS measurements are used to produce three long term data records, vertically resolved profiles of ozone, nitrogen dioxide and sulphate aerosols in the stratosphere and upper troposphere, and recently these processing chains have been merged, resulting in data product Version 7.2 for each of the three species. Details related to the sulphate aerosol processing and the nitrogen dioxide processing can be found in Rieger et al. (2019) and Dubé et al. (2022) respectively. As the OSIRIS ozone retrieval is now coupled to the retrievals of these related species, they have a small impact on the OSIRIS ozone data record (Bognar et al., 2022). This work by Bognar and co-authors also details the changes between the previous version (V5.10) and Version 7.2 of the OSIRIS ozone time series.

Although the impact was minor, within the Version 7.2 update many small changes were introduced to the OSIRIS data processing. The Multiplicative Algebraic Reconstruction Technique (MART) (Degenstein et al., 2009) has been replaced by a Levenberg-Marquardt scheme; the OSIRIS pointing correction (Bourassa et al., 2018) developed specifically for the retrieval of ozone has been implemented for all species (nitrogen dioxide and sulphate aerosols) further minimizing the impact on ozone of errors in these related species retrievals; the temperature dependent OSIRIS spectral response function in the 320 nm region of the measured spectra has been diagnosed and a correction has been implemented (see Appendix A of Bognar et al., 2022); the standard OSIRIS processing now uses temperature and pressure fields from the Modern-Era Retrospective analysis for Research and Applications – 2 (MERRA-2) as described in Wargan et al. (2017) and Gelaro et al. (2017); the BDM ozone absorption cross section, named for the authors of (Brion et al., 1993; Daumont et al., 1992; Malicet et al., 1995), has been implemented as part of the ozone retrieval; and finally the lower bound of the OSIRIS ozone retrievals has been better defined through use of a new cloud detection scheme implemented for the OSIRIS aerosol product and described with Rieger et al., 2019. For more complete documentation of the OSIRIS version 7.2 ozone data product please refer to the following web page (https://arg.usask.ca/docs/osiris_v7/index.html, last access: 09 October 2022).

## 2.5    OMPS-LP ozone profiles processed by University of Bremen

To retrieve the vertical distribution of ozone in the stratosphere from OMPS-LP observations the radiative transfer model SCIATRAN is used, with a Tikhonov regularization approach to constrain the profile. Four spectral segments are selected: three in the UV region and one in the Chappuis band. The altitude range over which the retrieval is performed spans between 8 and 60 km above sea level. Limb radiance in each spectral interval is first normalized with respect to a limb measurement at an upper tangent height. Simultaneously with the ozone retrieval, surface reflectance estimation is performed exploiting the



sun normalized radiance at 38–40 km, in the 340–345 nm and 675 nm ranges respectively. Beforehand, a cloud filter is applied and the retrieval of aerosol extinction profiles is performed. In addition, a retrieval of polar mesospheric cloud (PMC) properties is implemented and this information is used in the ozone retrieval in the presence of PMCs. This improves the coverage of the data set in polar regions during local summer. For a detailed description of the retrieval scheme see Arosio et

al. (2018). In the same paper, validation activities are also described, which mainly include comparisons with collocated ozonesondes and MLS observations. The discrepancies with respect to MLS profiles are well within ±10% between 20 and 58 km at all latitudes, whereas the disagreement increases in the tropical UTLS. Ozonesondes are used to validate these OMPS-LP data in the lower stratosphere and the best agreement was found at northern mid-latitudes with average discrepancies within ±3 % between 12 and 28 km. In the tropics, an average discrepancy of about 8–12 % is found between 15 and 19 km. In the

Southern Hemisphere the comparison is not as good as in the Northern Hemisphere, but still within ±7 % below 30 km.
The typical vertical resolution of the retrieved profiles is about 2–3 km, with larger values in the 30–35 km range, below 20 km in the tropics and above 50 km at all latitudes. A thorough uncertainty analysis was performed and presented in Arosio et al. (2022). The typical retrieval noise spans 2–3% between 15 and 50 km. The total random uncertainty is estimated in the range 3–5 % in the middle stratosphere increasing in the UTLS. The total systematic uncertainty is mainly related to cloud

contamination and model errors in the lower stratosphere, and to the retrieval bias at high altitudes, with total absolute values of about 5 % above 50 km and below 20 km.

## 2.6    SAGE III/ISS v5.2

The second instrument of the SAGE III project (Chu and Veiga, 1998) was launched to the ISS in February 2017 and began routine operations in June 2017 that continue to the present. In a mid-inclination (~52°) low Earth orbit (~420 km), SAGE

III/ISS (Cisewski et al., 2014) uses the solar occultation technique (McCormick et al., 1979) to make vertical profile measurements of ozone, aerosol extinction, water vapor, and nitrogen dioxide that cover the ~70°S–70°N range on a monthly basis. While the instrument also makes measurements using the techniques of lunar occultation and limb scattering, those data are not used here. A detailed description of the solar occultation ozone retrieval can be found in Wang et al. (2020). Briefly, the "AO3" ozone product that is used here, is derived from measurements in the Chappuis band (near 600 nm) simultaneously

with aerosol using measurements made in select channels across the visible and near infrared range (~520–1020 nm). This ozone product is reported from the surface or cloud top up to 70 km on a 0.5 km grid with a vertical resolution of ~1 km. The "AO3" ozone product is preferred over the other stratospheric ozone product (i.e., "MLR") because it has the best precision (~2–4%) and smallest differences compared to other satellites and ground stations (<5% throughout the stratosphere (Wang et al., 2020). The version of data used here is v5.2, which is produced by NASA, and is available at

https://eosweb.larc.nasa.gov/project/SAGE%20III-ISS, but it is also routinely transformed to the HARMOZ data format in the CCI project.





### 2.7    POAM III on SPOT 4

POAM III is a solar occultation instrument on the SPOT 4 satellite, which operated from 1998 to 2005. It flew in a sun-synchronous polar orbit, performing solar occultation observations in nine band channels, covering the spectral range from 354 to 1018 nm. Successive measurements cover a fairly constant latitude band (55–71° N for sunrise, 63–88° S for sunset).

The altitude range for the NASA v4 retrieved ozone profiles spans the 5–60 km range, with a vertical resolution of 1 km in the stratosphere and 2–5 km in the upper troposphere. Typical retrieval errors are reported to be within 5% in the stratosphere and increasing up to 15–30 % in the troposphere. Aerosol extinction and sunspots are known to affect the retrievals from POAM III observations, mainly in the 20–40 km altitude range. However, according to Lumpe et al. (2002),  less than 10% of the ozone profiles are reported to suffer from sunspot-related artifacts, which results in random errors of more than about 10% in the stratosphere.

The POAM III ozone product was validated against SAGE II, HALOE and balloon-borne observations, as described in Lumpe et al. (2002) and Randall et al. (2003). Results showed an agreement within 5% in the 13–50 km range, whereas a 15–20% high bias was found at lower altitudes.

### 2.8    Other datasets

In various data intercomparisons, we also used MLS v.4.2 ozone profiles (MLS data were also transformed to the HARMOZ format), as well as HEGIFTOM and SHADOZ ozonesonde profiles (available at https://hegiftom.meteo.be/datasets/ozonesondes and https://tropo.gsfc.nasa.gov/shadoz/Archive.html, Witte et al., 2017; Thompson et al., 2017; Witte et al., 2018; Sterling et al., 2018; Van Malderen et al., 2016).  For the comparison with OMPS-LP data, ozonesonde profiles were collocated with the satellite observations and the OMPS-LP averaging kernels were applied to degrade the high vertical resolution of sondes.

### 3    Data merging for the SAGE-CCI-OMPS dataset and its updated version

In this section, we present a short description of the merging method for the SAGE-CCI-OMPS dataset, and its modification for the SAGE-CCI-OMPS+ dataset. We also present detailed comparisons of deseasonalized anomalies for new datasets included in SAGE-CCI-OMPS+.

### 3.1    A short description of the merging algorithm for the SAGE-CCI-OMPS dataset

A detailed description of the merging algorithm used for the SAGE-CCI-OMPS dataset is presented in Sofieva et al. (2017). Here we present a short description of this algorithm.


For the merged dataset, first the monthly zonal mean ozone profiles in 10° latitude bands from individual instruments are computed. Then, for each instrument, the deseasonalized anomalies are computed as

$$\Delta(t_i) = \frac{\rho(t_i) - \rho_m}{\rho_m} \, ,$$
(1)

where $\rho(t_i)$ is the monthly zonal mean number density value at a certain altitude and latitude band corresponding to time $t_i$

and $\rho_m$ is the mean over the whole considered time period, for the corresponding month $m$ for this altitude and latitude. For GOMOS, MIPAS, SCIAMACHY, ACE-FTS and OSIRIS, the seasonal cycle is evaluated using the overlapping period 2005–2011. The seasonal cycle for SAGE II is computed using years 1985–2004 and for OMPS using the years 2012–2020.

The merging is performed as follows. In the first step, the median of GOMOS, MIPAS, SCIAMACHY, ACE-FTS and OSIRIS deseasonalized anomalies is computed (pre-merging). In the second step, SAGE II deseasonalized anomalies are

offset to the pre-merged anomalies in the years 2002–2005. The OMPS deseasonalized anomalies are offset to pre-merged anomalies (which are based on OSIRIS and ACE-FTS measurements in this period) in the years 2012–2020. After offsetting, all deseasonalized anomalies are aligned and the median of deseasonalized anomalies from all instruments is computed. In the merging, we also applied a method for detection of outliers: we filtered out individual anomaly values (locally for each latitude band and altitude level), which differ from the median anomaly more than 10% at latitudes 40°S–40°N and more than 20% in

other latitude bands. These thresholds are rather loose and do not affect the merged ozone in the overwhelming majority of cases; it removes only a few exceptional anomalies, which appear in rare cases for the instruments with rather coarse sampling (such as GOMOS and ACE-FTS).

The merged deseasonalized anomalies can be used directly to estimate ozone trends. For other applications, the merged ozone number density profiles are also provided. The computation of number density profiles from the merged

deseasonalized anomalies is performed via restoring the seasonal cycle according to Eq.(1). For the SAGE-CCI-OMPS, the amplitude of the seasonal cycle is estimated using MIPAS measurements because they provide all season pole-to-pole measurements with dense sampling. The absolute values of the seasonal cycle are estimated from SAGE II and OSIRIS in the overlapping period (which are very close to each other and to GOMOS measurements), thus preserving the consistency in the dataset through the whole observation period.

### 3.2    Comparisons of the new datasets

Different intercomparisons of the new datasets have been performed. In this section, we show some illustrations and discuss the impacts of using new datasets.

On the monthly zonal mean level, MIPAS v8 ozone profiles are 1–2% larger than those of v7 in the middle stratosphere, and ~1 % smaller at ~45 km, as illustrated in Figure 1. This decrease at ~45 km seems to be related to a change

of retrieval grid width at this altitude in v7, while the v8 processor uses the grid with a constant spacing up to 55 km. In the



mesosphere, v8 reports 2–6 % smaller values over the tropics and middle latitudes. In the polar winter stratosphere, MIPAS v8 has 2–7 % larger values compared to v7 data. The differences in the UTLS are of larger magnitude (~10 %) and they change from altitude to altitude (Figure 1). Figure 2 shows the difference in deseasonalized anomalies (defined by Eq.(1)). As illustrated in Figure 2, the deseasonalized anomalies computed from MIPAS v8 and v7 data are very similar (typical patterns

of the anomalies themselves can be seen in Figure 12, they are in the range of ~±20% ).

ACE-FTC v4.1/4.2 monthly zonal mean ozone profiles are 2-4% larger than those of v3.5/3.6 in the middle stratosphere, and ~1−4 % smaller at altitudes 50−60 km, as illustrated by Figure 3. The differences in the UTLS are ± 4−8 %, their sign and magnitude are altitude and latitude dependent. ACE-FTS v.4.1/4.2 ozone data have a smaller drift. This is illustrated in Figure 4, which compares differences to MLS deseasonalized anomalies at 40 km for ACE-FTS v3.5/3.6 ozone

profiles (panel a) and ACE-FTS v.4.1/4.2 data (panel b). The progressing differences (drift) of v3.5/3.6 with respect to MLS largely disappears for v.4.1/4.2. This deviation of ACE-FTS v3.5/3.6 with respect to MLS anomalies increases with altitude between 10–50 km (not shown). The analogous behavior is also observed in comparison with the merged SAGE-CCI-OMPS deseasonalized anomalies (Figure S1 in the Supplement), but the difference is slightly less visible, because ACE-FTS data are used in construction of SAGE-CCI-OMPS merged dataset.

OSIRIS v7.2 ozone profiles cover lager altitude range, compared to the OSIRIS v.5.10 data. This can be seen by comparing panels (b) and (a) in Figure 5. OSIRIS v7.2 monthly zonal mean ozone profiles are 1−5 % smaller in the middle stratosphere, mostly ~2−7 % larger in the lower stratosphere, and more than 10 % smaller in the troposphere, compared to analogous monthly zonal mean ozone profiles evaluated using OSIRIS v5.10 data. The difference with respect to the analogous MLS deseasonalized anomalies are slightly different between OSIRIS v.5.10 and OSIRIS v.7.2: the latter anomalies are closer

to those of MLS before 2006, but larger in 2014−2016, as shown in Figure 6.



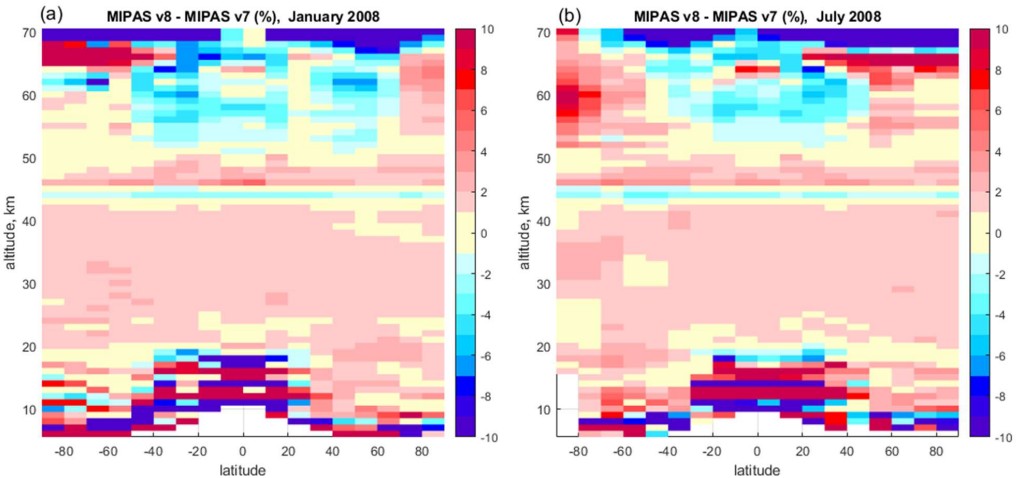

**Figure 1. Relative difference of MIPAS v8 and v7 monthly zonal mean ozone profiles for January 2008 (panel a) and July 2008 (panel b).**

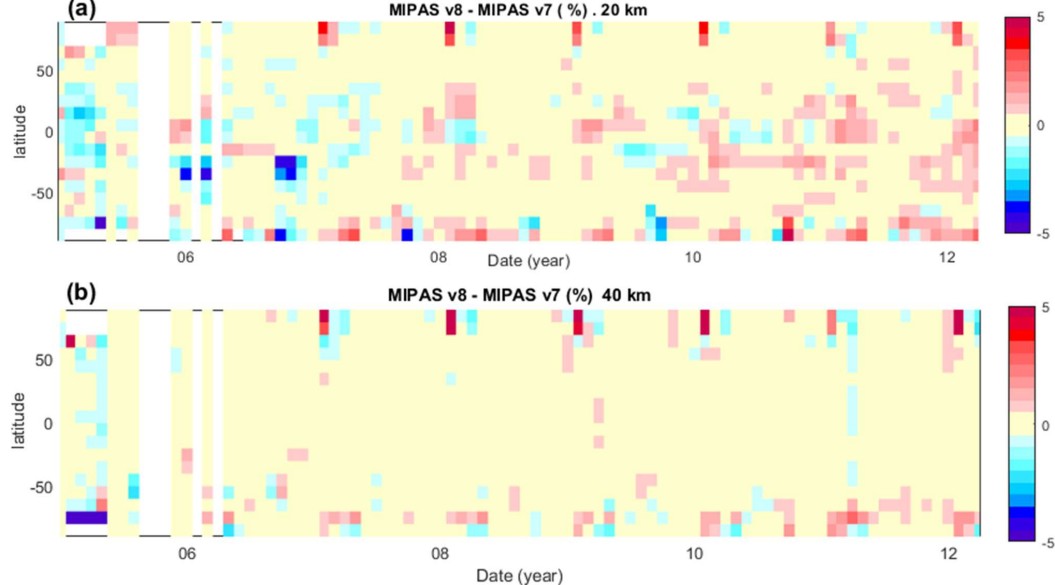

5    **Figure 2.  Difference in the deseasonalized anomalies between  MIPAS v8 and v7  ozone data (v8 minus v7) in %,  for altitudes 20 km (panel a) and 40 km (panel b).**





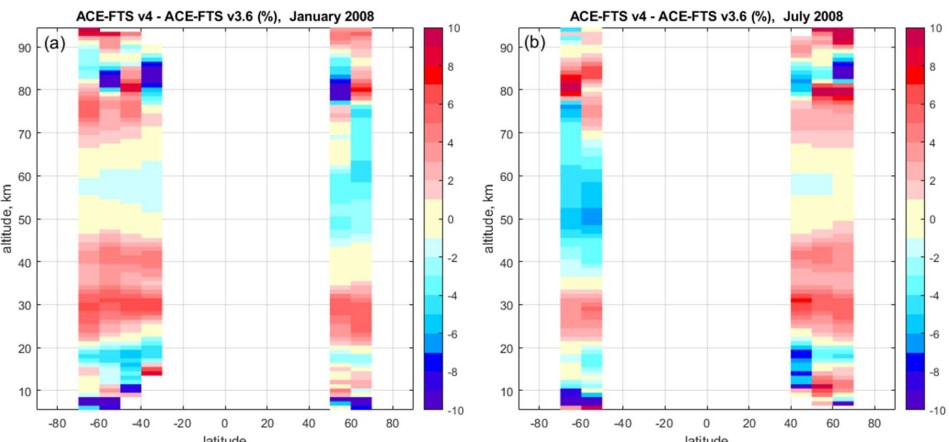

**Figure 3.** Relative difference of ACE-FTS v4.1/4.2 and v3.5/3.6 monthly zonal mean ozone profiles for January 2008 (panel a) and July 2008 (panel b)

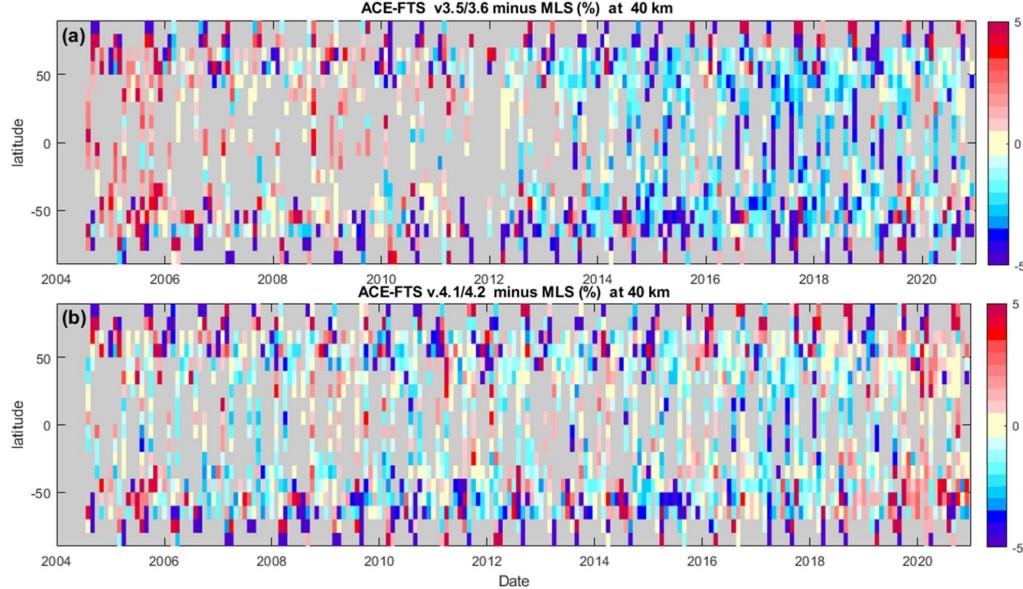

**Figure 4.** Difference of deseasonalized anomalies ACE-FTS minus MLS in % at 40 km, for ACE-FTS version 3.5/3.6 (panel a) and version 4.1/4.2 (panel b).



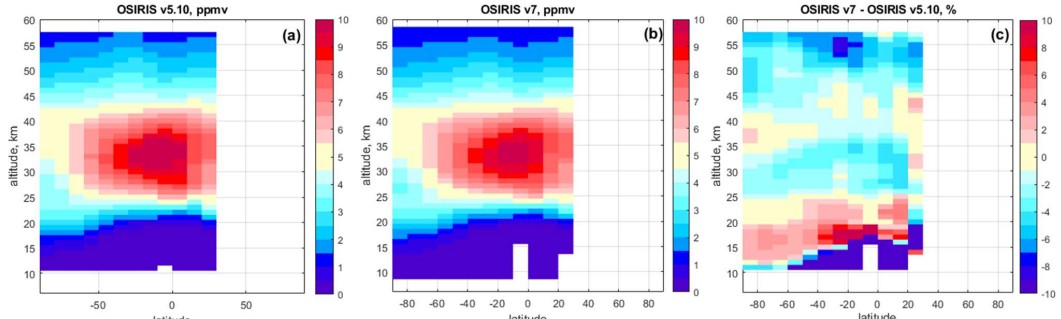

**Figure 5. Monthly zonal mean ozone profiles in January 2008 evaluated using OSIRIS v5.10 data (panel a), OSIRIS v.7 data (panel b). Panel (c): relative difference of OSIRIS v7 and v5.10 monthly zonal mean ozone profiles for January 2008.**

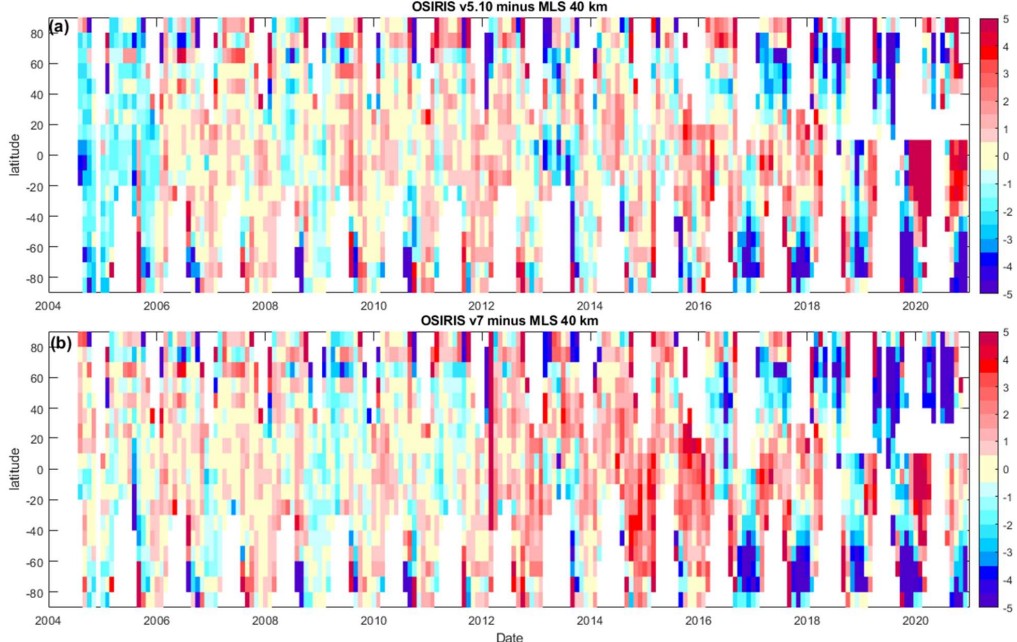

**Figure 6 Difference of deseasonalized anomalies OSIRIS minus MLS at 40 km, for OSIRIS version 5.10 (panel a) and version 7 (panel b).**



The OMPS-LP ozone profiles processed by University of Bremen and University of Saskatchewan have slightly different vertical extent (larger for OMPS UBr), as illustrated in Figure 7 (a, b). Due to different thresholds on solar zenith angle, polar regions are better covered by OMPS USask data (Figure 7 a, b). The biases of monthly zonal mean data can be up to 10 % in the middle stratosphere and even larger near the tropical tropopause (Figure 7c). Despite the difference in absolute ozone

values, the deseasonalized anomalies from OMPS UBr and USask data are very similar in majority of cases, as illustrated in Figure 8, which suggests the idea of using the mean of UBr and USask deseasonalized anomalies as the OMPS anomalies might be viable. The averaging can be done ignoring missing data (i.e., if data are missing in one of the datasets, it will follow the existing data from another dataset). The intercomparison of OMPS and UBr ozone deseasonalized anomalies is aimed at assessing whether averaging of deseasonalized anomalies is advantageous, as well as at defining a valid range of OMPS data.

For this, the OMPS UBr and USask deseasonalized anomalies are compared with deseasonalized anomalies from MLS and from the merged SAGE-CCI-OMPS dataset.

In polar regions, USask data have better coverage, as illustrated in Figure 9, which compares the deseasonalized anomalies from UBr (panel a), USask (panel b) and MLS (panel d) for 70–80°N. Using OMPS-LP USask data in cases when UBr data are missing is advantageous from the point of view of data coverage. In these cases, the OMPS USask anomalies are close to

those of MLS in the majority of cases. Here we would like to note that the data with strongly inhomogeneous spatial or temporal sampling are not used in the deseasonalized anomalies (see also Sect. 3.3). There are also periods when OMPS USask data have pronounced deviations from MLS anomalies that occur in winters with large anomaly values (e.g. in NH winter 2016 and 2018, see also Figure S2 in the Supplement), while these periods are not covered by UBr data. At lower altitudes, the mean of OMPS UBr and USask anomalies are closer to MLS anomalies, compared to each dataset separately, as observed in Figure S2

(Supplement). Such behavior – smaller deviations of the mean of the OMPS UBr and USask deseasonalized anomalies from MLS or from merged SAGE-CCI-anomalies – are also observed for other latitude bands as illustrated in Figure 10. This indicates that averaging of OMPS UBr and USask deseasonalized anomalies is advantageous.

In the tropical upper stratosphere at 14–16 km, we found a strong drift (or a step) in OMPS UBr data, which was observed in comparison with MLS, merged SAGE-CCI-OMPS dataset and ozonesondes (illustrations can be found in the Supplement,

Figures S3–S6). At other altitudes in the tropical troposphere and UTLS, OMPS-LP UBr data show reasonable agreement with ozonesondes and MLS data: at altitudes 10-13 km, ozone variations and differences between different data are large, as illustrated in Figure S4, but the drift is strongly reduced, as can be seen in Figure S5. Comparing sondes, MLS and OMPS UBr at 15.5 km (Figure S6), we notice that the negative drift at this altitude with respect to MLS is enhanced by the positive drift of MLS with respect to sondes.






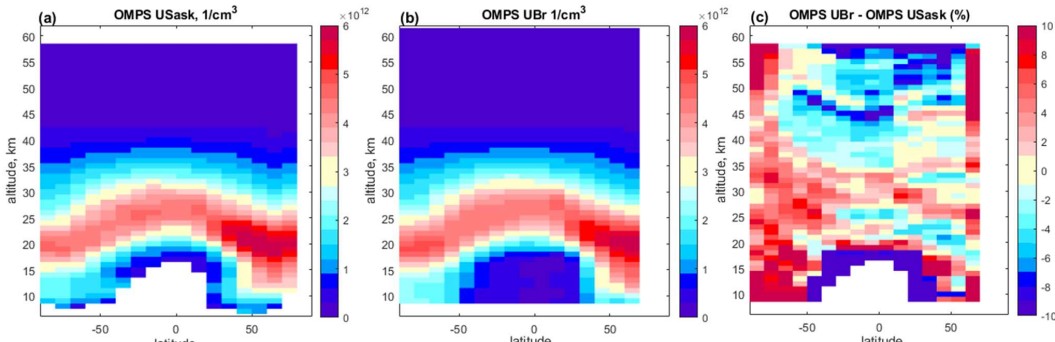

**Figure 7. Monthly zonal mean ozone profiles in January 2018 evaluated using OMPS-LP USask data (panel a), OMPS-LP USask data (panel b). Panel (c): relative difference of OMPS UBr and OMPS USask monthly zonal mean ozone profiles for January 2018.**

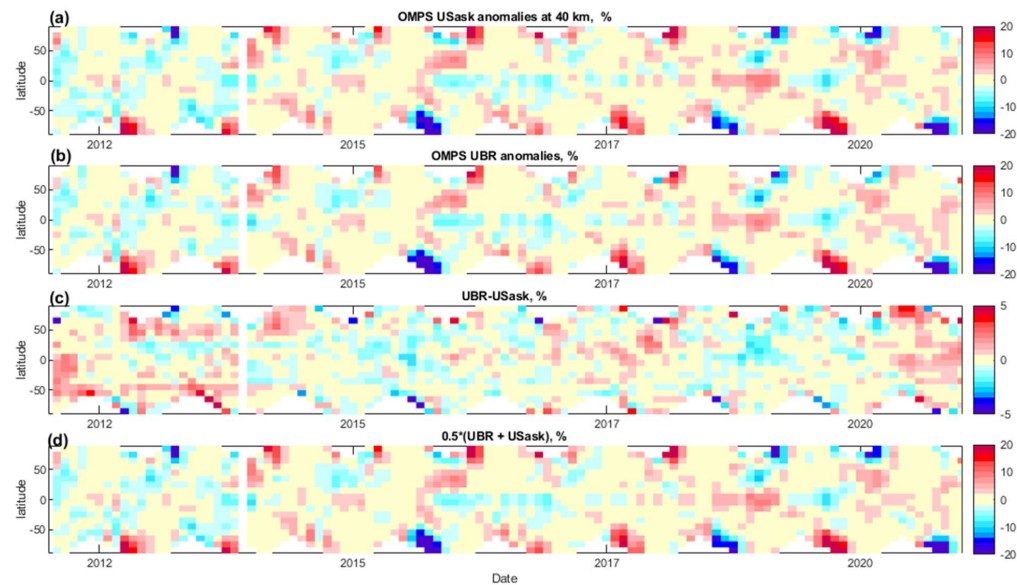

**Figure 8 (a): OMPS USask deseasonalized anomalies at 40 km, (b): OMPS UBr deseasonalized anomalies at 40 km, (c): Difference of OMPS UBr and USask deseasonalized anomalies at 40 km, (d): mean of OMPS UBr and USask deseasonalized anomalies at 40 km.**





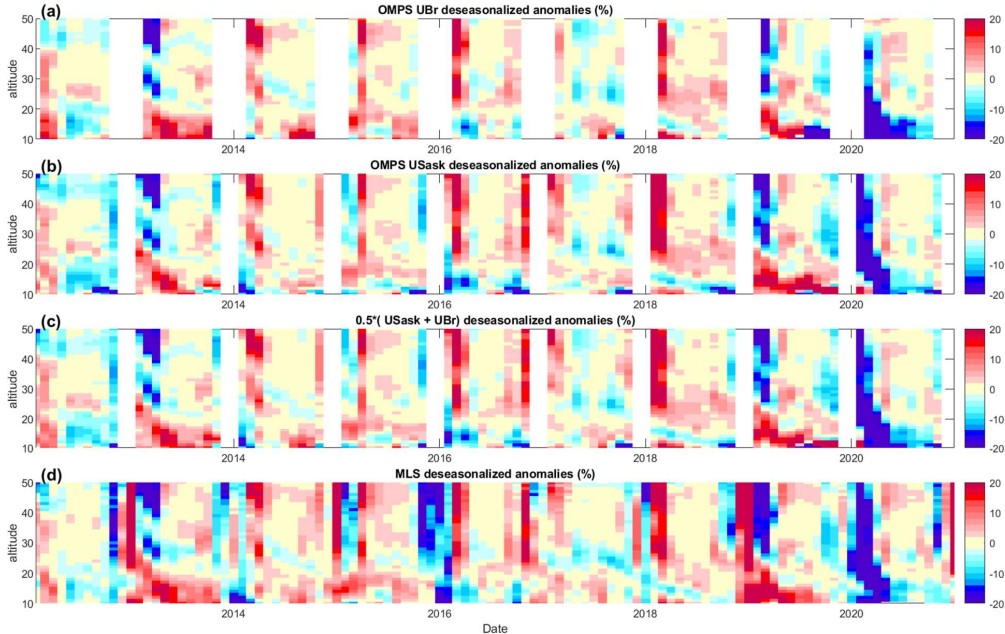

**Figure 9 Deseasonalized anomalies at 70-80 N for (a) OMPS UBr ozone data, (b) OMPS USask ozone data, (c) mean of of USask and UBr deseasonalized anomalies (ignoring missing data), and (d) MLS deseasonalized anomalies.**





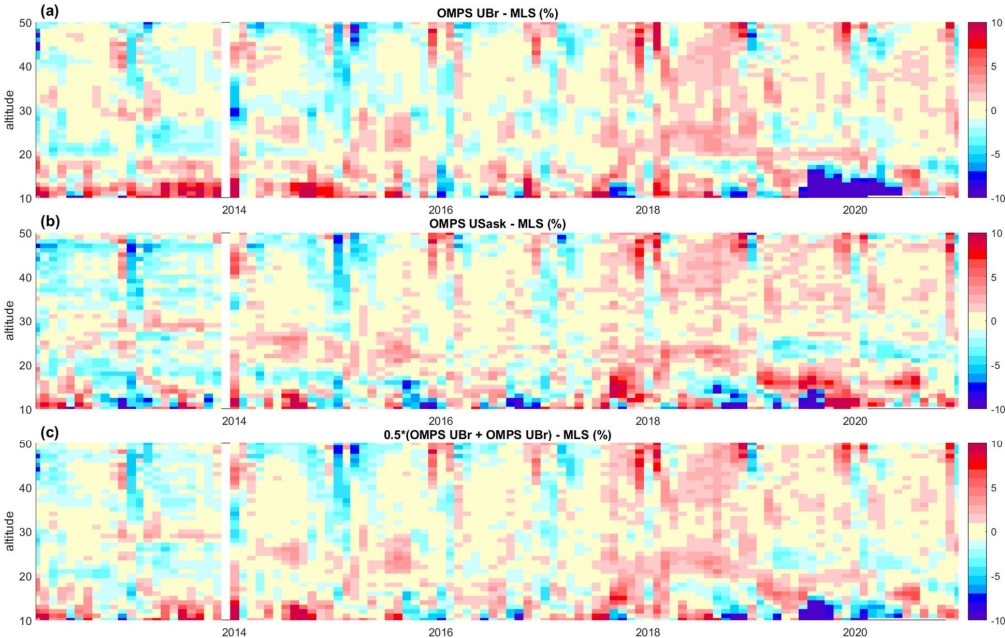

**Figure 10. Difference between OMPS and MLS deseasonalized anomalies at 50-60 N for (a) OMPS UBr, (b) OMPS USask, and (c) mean of OMPS UBr and USask datasets.**

5     POAM III data mainly cover the polar regions in 1998–2005, while SAGE III/ISS are mostly within ~60°S–60°N latitude range (Figure 11). As observed from Figure 11, which shows the difference of POAM III ad SAGE III/ISS deseasonalized anomalies with respect to the merged SAGE-CCI-OMPS anomalies, the anomalies are in rather good agreement with the merged SAGE-CCI–OMPS dataset at a majority of altitudes. One can notice a larger deviation of SAGE III/ISS from the baseline SAGE-CCI–OMPS anomalies at 20 km in the tropics over the first year of SAGE III/ISS data (Figure 11). In

10    comparison with MLS, however, the same deviation in SAGE III/ISS data is only noticeable between Dec 2017 – Jan 2018, albeit with a smaller magnitude (Figure S7 in the Supplement). At the moment, the SAGE III/ISS observational period is too short to make conclusions about the systematicity of deviations.



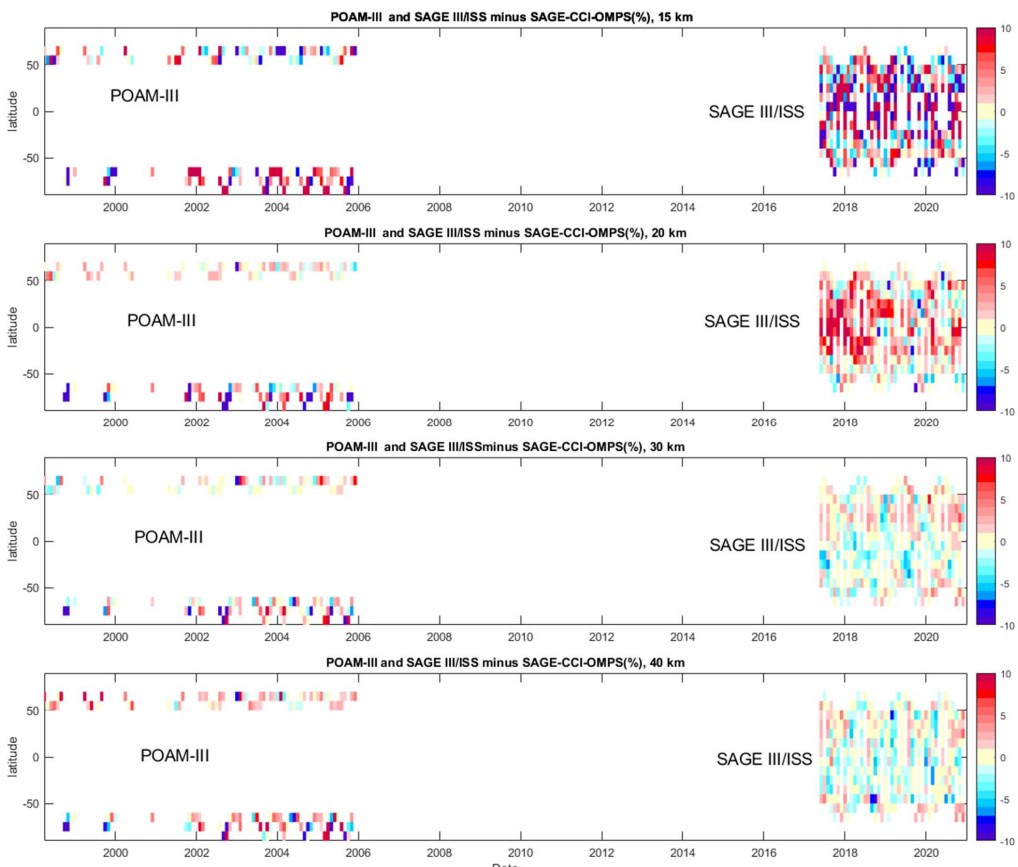

**Figure 11.** Difference of deseasonalized anomalies POAM III and SAGE III/ISS and the merged SAGE-CCI-OMPS deseasonalized anomalies, at several altitude levels.

## 3.3 Data merging in SAGE-CCI-OMPS+ dataset

The merging procedure of SAGE-CCI-OMPS+ dataset is very similar to that for SAGE-CCI-OMPS dataset, it is based on using the deseasonalized anomalies. Compared to SAGE-CCI-OMPS dataset, we improved the data filtering from latitude-time bins with highly inhomogeneous sampling: now they are ignored if either inhomogeneity measure in latitude $H_{lat}$ or in time $H_{time}$ exceed 0.9 (the definitions of the inhomogeneity measure is presented in Sofieva et al. (2017, 2014). This filtering



only removes a few data points corresponding to highly inhomogeneous sampling, which are mostly for occultation instruments and for data in polar regions.

Similarly to the SAGE-CCI-OMPS dataset, we use only the data, which are in good agreement and do not exhibit significant drifts with respect to collocated ground-based observations and with respect to each other. For this reason, we do

not to use the SCIAMACHY data before August 2003, OMPS data before April 2012, GOMOS data after November 2011, and MIPAS data in 2002–2004 in the merged dataset (the illustrations and discussions are presented in Sofieva et al. (2017). As mentioned above, in SAGE-CCI-OMPS+ we excluded OMPS UBr data at altitudes 14–16 km at latitudes 20°S–20°N. We use the mean of OMPS USask and UBr deseasonalized anomalies (ignoring missing values) as the contribution from OMPS.

Using the same technique as for the SAGE-CCI-OMPS dataset, we perform first pre-merging via computation of the

median GOMOS, MIPAS, SCIAMACHY, ACE-FTS and OSIRIS deseasonalized anomalies. Then SAGE II, OMPS, POAM III and SAGE III/ISS deseasonalized anomalies are offset to the pre-merged anomalies using the corresponding overlapping periods. After that, the outlier detection and the final merging (computing the median of all aligned deseasonalized anomalies) is performed. As an example, Figure 12 shows deseasonalized anomalies from individual instruments and merged anomalies at 30 km.

The procedure of reconstruction of ozone concentrations from the merged deseasonalized anomalies and error estimations is the same as for the SAGE-CCI-OMPS dataset. It is described in Sect. 3.1. Some illustrations of SAGE-CCI-OMPS and SAGE-CCI-OMPS+ datasets are shown in Figure 13, which compares ozone time series at 30 km and ozone profile time series close to the Equator (0–10°N). As shown in Figure 13(a,c), the SAGE-CCI-OMPS+ dataset has a better coverage of polar regions due to inclusion of POAM III data. The SAGE-CCI-OMPS+ dataset has also a better coverage of the UTLS

region (Figure 13b,d) due to inclusion of SAGE III and OMPS-LP UBr data.

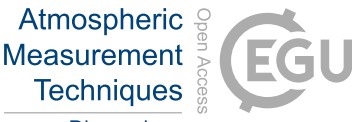

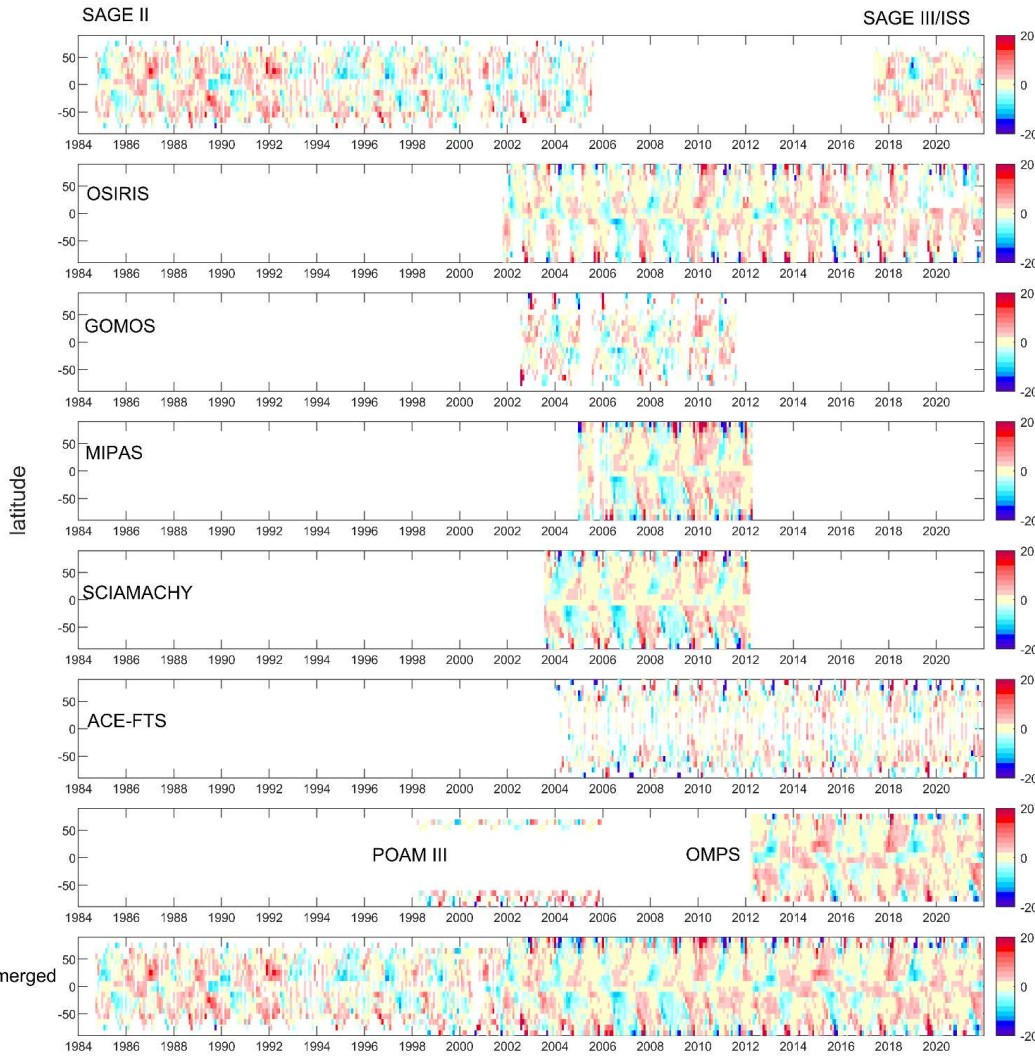

**Figure 12.** Deseasonalized anomalies from individual datasets and merged at 30 km





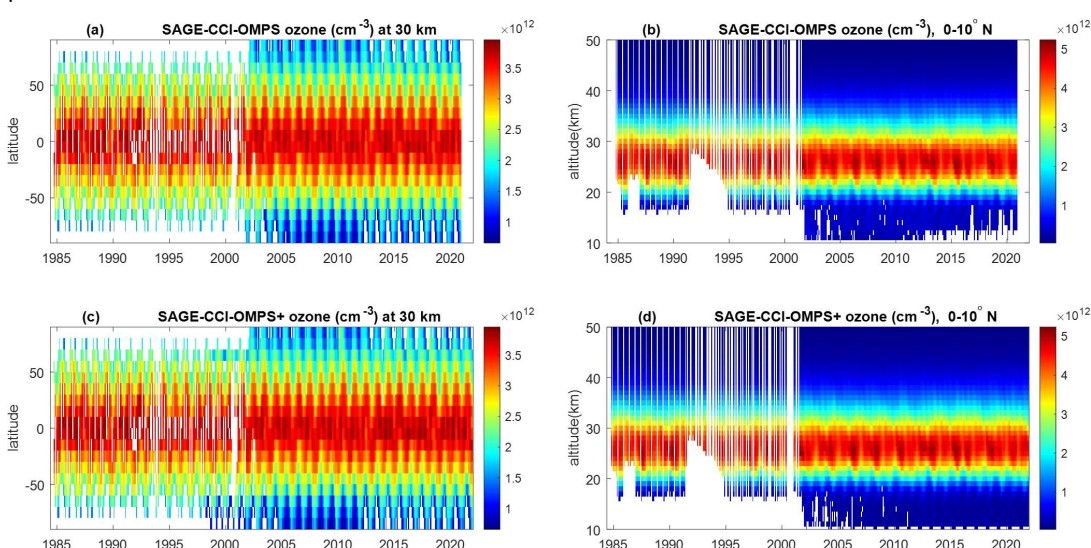

**Figure 13. Time series of ozone at 30 km in the merged SAGE-CCI-OMPS (panel a) and SAGE-CCI-OMPS+ (panel c) datasets.
Time series of ozone profiles at 0–10°N in SAGE-CCI-OMPS (panel b) and SAGE-CCI-OMPS+ (panel d).**

## 4   Sensitivity of trend analysis to the inclusion of new datasets

The objective of the analyses presented in this section is the investigation of sensitivity of ozone trends to the changes in the
datasets used for data merging. For this, we created the series of test datasets, in which only one change was introduced. For
example, MIPAS v7 was replaced by MIPAS v8, or ACE-FTS v3.5/3.6 was replaced by ACE-FTS v4.1/4.2, added SAGE III
10   while other datasets are as in SAGE-CCI-OMPS dataset, and so on.

For studies of trends sensitivity, we used a simple regression model

$$O_3(t) = PWLT(t,t_0) + q_1 QBO_{30}(t) + q_2 QBO_{50}(t) + s\,F_{10.7}(t) + d\,ENSO(t),\qquad (2),$$

where $PWLT(t, t_0)$ is a piecewise linear term (constant and a hockey-stick trend with the turnaround point in 1997), $QBO_{30}(t)$
15   and $QBO_{50}(t)$ are the equatorial winds at 30 hPa and 50 hPa, respectively (http://www.cpc.ncep.noaa.gov/data/indices/), $F_{10.7}(t)$
is the monthly average solar 10.7 cm radio flux (ftp://ftp.geolab.nrcan.gc.ca/data/solar_flux/monthly_averages/), and $ENSO(t)$
is the 2 month lagged ENSO proxy (http://www.esrl.noaa.gov/psd/enso/mei.table.html).



Although the regression model is rather simple, it was used in previous trend analyses (Kyrölä et al., 2013; Sofieva et al., 2017, 2021). It was shown in Petropavlovskikh et al. (2019) that the evaluated trends weakly depend on the regression model. We would like to add a caveat that the trend estimates in polar regions are less accurate, due to the large year-to-year variability and absence of dynamical proxies (which can partly explain this variability) in the regression model.

Figure 14 and Figures S7–S12 in the Supplement show the differences in trend estimates caused by changes in the dataset. The figures in the Supplement show the trends for the baseline SAGE-CCI-OMPS dataset (left panels), trends when the corresponding change in one dataset was introduced (central panels), and the difference (new minus baseline), which are shown in right panels of Figures S7–S12 and also in Figure 14.

Using ACE-FTS v4.1/4.2 instead of ACE-FTS v3.5/3.6 results in ~0.5–0.7% decade$^{-1}$ larger ozone trends in the upper
stratosphere, as shown in Figure 14(a) and Figure S8 (consistently with analyses presented in Sect.3.2.). Using MIPAS v8 instead of MIPAS v7 affects the trend estimates only at high latitudes (Figure 14(b) and Figure S9), where the estimated trends become ~1% decade$^{-1}$ larger. Using the new OSIRIS ozone dataset version 7 affects the trends at all altitudes and all latitudes: ozone trends become ~0.5–1% decade$^{-1}$ larger in the middle stratosphere, but ~0.5–1% decade$^{-1}$ smaller in the northern upper stratosphere (Figure 14(c) and Figure S10). Using both OMPS USask and UBr datasets has a rather small impact on evaluated
ozone trends; the visible reductions are in the tropical upper troposphere and close to North Pole (Figure 14(d) and Figure S11).  Adding SAGE III data has a small impact on ozone trends (Figure 14(e) and Figure S12). This is rather expected, as SAGE III deseasonalized anomalies agree very well with the merged SAGE-CCI-OMPS dataset, and the SAGE III time series is relatively short (Sect 3.2.). Adding ozone profiles from POAM III results in a slight decrease of ozone trends at 60–80°S (by ~0.3–1 % decade$^{-1}$ ), and a pronounced increase of ozone trends, more than 1% decade$^{-1}$, near the South Pole (Figure 14(f)
and Figure S13).

Overall, using new datasets has a rather small impact on the resulting trends at latitudes 60°S–60°N (typically within ±0.5 % decade$^{-1}$). The effect is more pronounced in polar regions, which is rather expected due to scarcer data coverage, and significantly larger ozone variability (note the caveats on the regression model mentioned above).



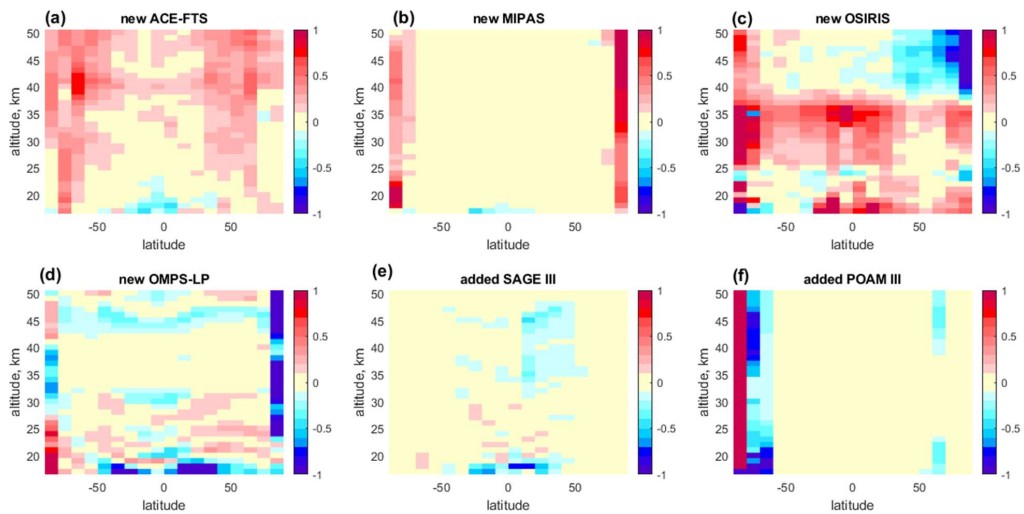

**Figure 14 Difference in trend estimated (in % dec⁻¹) cause by changes in the datasets (new minus baseline, see text for explanation)**

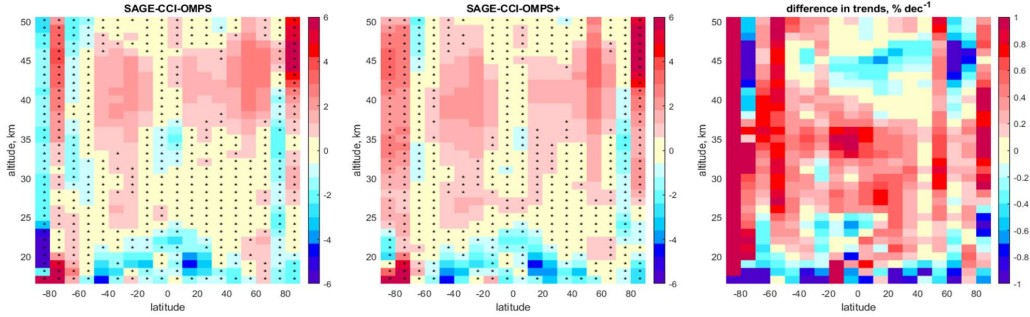

5 **Figure 15. Ozone trends in 1997-2020 (% decade⁻¹) evaluated using the baseline SAGE-CCI-OMPS dataset (left) and the SAGE-CCI-OMPS+ dataset. Stars indicate the latitude-altitude bins, in which trends are not statistically significant at 95% confidence level. Right panel shows the difference in ozone trends in % decade⁻¹ (new minus baseline)**

Post-1997 ozone trends evaluated using SAGE-CCI-OMPS and SAGE-CCI-OMPS+ datasets (the regression analysis

10 has been done using the common period 1984–2020) are shown in Figure 15. The overall structure of ozone trends is very

similar in both merged datasets: the trends in the upper stratosphere are positive and statistically significant. The largest





changes are at highest latitudes (80°–90°, especially in the Southern Hemisphere). At other latitude bands, the influence on ozone trends is rather small (usually less than 0.5 %/ dec).

## 5    Summary

SAGE-CCI-OMPS+ is an updated version of the SAGE-CCI-OMPS dataset. In addition to the ozone profiles from SAGE
II, OSIRIS, GOMOS, MIPAS, SCIAMACHY, ACE-FTS and OMPS-LP used in the original dataset, SAGE-CCI-OMPS+ also includes the ozone profile datasets from POAM III and SAGE III/ISS. For MIPAS, ACE-FTS and OSIRIS, the ozone data from updated processors are used. In the updated dataset, both OMPS-LP ozone profiles processed by University of Saskatchewan and University of Bremen are used; taking the mean of deseasonalized anomalies as an OMPS-LP dataset improves the spatial coverage and agreement with other datasets. We performed detailed intercomparison of datasets from
individual instruments. The new processed ozone datasets from ACE-FTS, MIPAS and OSIRIS are expected to be more stable.

The merging method is similar to that used for creating the SAGE-CCI-OMPS dataset: it is based on the median of aligned deseasonalized anomalies from individual instruments.  The updated SAGE-CCI-OMPS+ has a better coverage of polar regions and the UTLS.

We analyzed sensitivity of ozone trends, which are estimated using multiple linear regression, to inclusion of new
datasets. Overall, the changes of ozone trends are within ±0.5% decade$^{-1}$ in the majority of latitude bins and altitudes, and they do not change the overall the morphology of trends in ozone profiles: the statistically significant trends are observed in the upper stratosphere.

The updated SAGE-CCI-OMPS+ dataset covers the period from October 1984 to December 2021, and it will be regularly extended in the future.  The profiles and ozone concentrations and deseasonalized anomalies are presented on altitude
grid from 10 to 50 km, and in 10° latitude bins from 90°S to 90°N.  The SAGE-CCI-OMPS+ dataset can be used for evaluation of ozone trends in the stratosphere and other research.

**Data availability**
The CCI datasets are available through open access at https://climate.esa.int/en/projects/ozone/data/ and at ftp://cci_
web@ftp-ae.oma.be/esacci (ESA Climate Office, last access: 26 August 2022).  The SAGE-CCI-OMPS dataset is available also through the Climate Data Store (CDS) of the Copernicus Climate Change Service (C3S) (https://cds.climate.copernicus.eu/cdsapp#!/dataset/satellite-ozone-v1?tab=overview).

**Acknowledgements**
The work was performed in the framework of the ESA Ozone_cci+ project (contract No. 4000126562/19/I-NB).  The KIT team would like to thank the European Space Agency (ESA) for giving access to MIPAS Level-1 data. The ozone retrieval from SCIAMACHY and OMPS-LP instruments at the University of Bremen was funded in parts by ESA (including the Living





Planet Fellowship SOLVE), German Aerospace Agency (DLR), German Research Foundation (DFG) through the research unit VolImpact (grant no. FOR2820), University and State of Bremen. The data set was calculated with resources provided by the North-German Supercomputing Alliance (HLRN). The GALAHAD Fortran Library was employed in the retrieval scheme. The GOMOS ALGOM2s dataset was created in the framework of ESA ALGOM project. Processing of MIPAS data used in this study was partly funded by DLR under contract no. 50EE1547 (SEREMISA). The MIPAS dataset computations were performed in the frame of a Bundesprojekt (grant MIPAS_V7) on the Cray XC40 "Hazel Hen" of the High-Performance Computing Center Stuttgart (HLRS) of the University of Stuttgart. The FMI team thanks the Academy of Finland (Centre of Excellence of Inverse Modelling and Imaging, decision 336798). The ACE mission is supported primarily by the Canadian Space Agency (CSA). Odin is a Swedish-led satellite project funded jointly by Sweden (SNSB), Canada (CSA), France (CNES), and Finland (Tekes). The authors thank Gabriele Stiller (KIT) for comments.

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
