# Peer review of "Updated merged SAGE - CCI- OMPS+ dataset for evaluation of ozone trends in the stratosphere"

_Atmospheric Measurement Techniques, 2022_

## Author Comment (AC2)

Dear reviewer,

Thank you very much for your positive comments on our paper. We took your comments into account in the revised version of the manuscript. Please find below our detailed replies (black font) on your comments (blue font).

**Minor Comments**:

When comparing the versions of the data for MIPAS and especially for ACE-FTS, I suggest cutting the figures off at 55km, or at least no higher than 60km. The ACE-FTS plots go up to 90km, which is far above the range of the merged record and thus not relevant to these results. Expanding the vertical scale will allow the authors to better highlight the regions that matter most to the merged product. In particular in ACE-FTS, there appears to be a seasonal component in the differences, with the mid-stratospheric increase from the old to new version over a broader vertical range in summer compared to winter.

In the revised version, we cut the figures at 55 km.

There also seems to be a bit of a trend relative to MLS in the new ACE-FTS version compared to MLS (Figure 4 bottom panel, also Fig. S1 but not as clear) starting in mid-2019 that might be worth noting as this would also contribute to a more positive trend.

We made this note in the revised version.

Page 18, Lines 15-20: For the details of reconstructing the merged record in absolute units (adding a seasonal cycle back in) and the error analysis, the readers are referred to Section 3.1 and the description of the original merged SAGE-CCI-OMPS. There is no discussion that I see concerning the error estimates, do the authors mean to refer to the 2017 paper? Did the inclusion of the new data lead to any notable changes in estimated uncertainties in the merged record? While the trend analysis including each instrument is very informative, a comment about any changes (or no changes) in uncertainty would be useful.

In the revised version, we added a short description of error estimation ( it is the same as for the SAGE-CCI-OMPS dataset). We also added a figure, which compares uncertainties of the original and the updated versions.

As for the seasonal cycle, section 3.1 says "For the SAGE-CCI-OMPS, the amplitude of the seasonal cycle is estimated using MIPAS measurements because they provide all season pole-to-pole measurements with dense sampling. The absolute values of the seasonal cycle are estimated from SAGE II and OSIRIS in the overlapping period (which are very close to each other and to GOMOS measurements), thus preserving the consistency in the dataset through the whole observation period." I'm not sure I follow the use of SAGE II and OSIRIS here, are SAGE II and OSIRIS seasonal cycles checked against the MIPAS seasonal cycle in their respective overlap periods with MIPAS, thus verifying using the MIPAS seasonal cycle over the full record is valid, or is the seasonal cycle from SAGE II and OSIRIS used directly? In any case, it appears there may be seasonal changes in some of the new data versions, an update to the seasonal cycles in Figure 4 of the 2017 paper would be useful as a supplemental figure that can be referred to in the text to support the representativeness of MIPAS to establish the seasonal cycle in the new version.

The best representation of the amplitude of seasonal cycle is provided by instruments with dense sampling and gap-free coverage. Therefore, we selected the MIPAS data for its evaluation.

As recommended, we included a figure illustrating the seasonal cycles in the Supplement.

When discussing trend results I assume all results are for the second portion of the piecewise linear fit (since 1997) as opposed to a linear fit over the full 'hockey-stick' proxy, but this should be specified in the text.

Yes, the trend results correspond to the second of the piece-wise linear fit. In the revised version, we indicate this explicitly.

**Typos/Editorial Suggestions**

Page 2 L2 The importance of monitoring stratospheric ozone and its vertical structure is well recognized …
L8 The main advantages of satellite …
L10 … instruments is limited, data from several instruments …
Page 3 L4 … ozone profiles are retrieved on a geometric altitude grid …
L5 presented on an altitude grid from 10 to 50 km.
L6 (upper troposphere and lower stratosphere)
L11 … we used ozone profile datasets …
L14 … we used altitude gridded datasets (HARMOZ_ALT), available …
L18 Add space before "Below"
Page 4 L9 described in Boone et al. (2020)

Corrected

L14: "Sheese et al. (2022) showed that v4.1 ozone data bias with respect to data sets" do the authors mean with respect to independent data sets?

Yes, we added "independent"

L23-24: This leads to less instrument drift in the retrieved ozone values.
Page 5 L6: OSIRIS measurements are used to produce three long term data records: vertically …
L7: … upper troposphere; recently these processing chains …
L21: authors of Brion et al. (1993), Daumont et al. (1992) and Malicet et al. (1995),
L23-24: described by Rieger et al. (2019).
L25: https://arg.usask.ca/docs/osiris_v7/index.html (last access: 09 October 2022).
Page 6 L24: the "AO3" ozone product used here is derived from measurements
Page 7 L6-7: within 5% in the stratosphere, increasing …

Corrected

L9: "which results in random errors of more than about 10%" do the authors mean 'less than' 10% here, or is 'more than' correct? It reads as though the less than 10% of ozone profiles suffer from sunspot-related artifacts should lead to lower random errors, if the more than 10% is correct, this should be re-worded.

We rephrased the text.

Page 8 L30: "retrieval grid width" consider adding vertical for clarity… retrieval vertical grid width
Page 9 L2: "(~10 %)" suggest changing to (~ +/- 10%) to clarify positive and negative range
L6: ACE-FTC -> ACE-FTS
L15: ozone profiles cover a larger altitude range
Page 14 Fig. 7 Caption: typo - Panel B is OMPS UBr
Page 16 L5: within the ~60°S–60°N latitude range or within~60°S–60°N latitude (remove range)
L6: POAM III and SAGE III/ISS

Corrected

Page 18 L3: How is good agreement defined? Consider re-wording as We use only the data that do not exhibit significant offset or drift with respect to …
L6: consider (illustrations and discussion of these data exclusions are presented in Sofieva et al. (2017)).

Reworded as suggested.

Page 21 L5 and L8: Figures S7-S12 should be Figures S8-S13
L10: (consistent with … )
Page 22 Fig. 15 caption: at the 95% confidence level
Page 23 L16: do not change the overall morphology of trends in ozone profiles: statistically significant trends …
L19: The profiles of ozone concentrations and deseasonalized anomalies are presented on an altitude grid …
Page 24 L4: framework of the ESA ALGOM project
Figure S3 caption: NLS -> MLS
Figure S5 and S6: specify this is OMPS UBr in the figure or caption

Corrected.

---

## Author Comment (AC3)

**Reviewer #3**

Dear reviewer,

Thank you very much for your positive comments on our paper. We took your comments into account in the revised version of the manuscript. Please find below our detailed replies (black font) on your comments (blue font).

**Reviewer#3 comments**

Page 3, line 11 – "ozone profiles datasets" -> "ozone profile datasets"

Corrected

Table 1: I suggest including a column with the reference to the appropriate instrument paper for the version of the data you are using for each instrument. Also, can you please check the vertical resolution numbers here? Saying these instruments have ~1 km vertical resolution seems awfully optimistic. Just because the vertical retrieval grid is at 1 km doesn't mean the vertical resolution is, as I'm sure the authors are aware. For example, I believe the OMPS vertical resolution should be more like 2-3 km, not ~1 km.

In the revised version, we added references to the instrument papers and revised the vertical resolution numbers.

Page 3, line 16: I was originally confused by why you were mentioning both the UBr and USask retrievals. The reasoning for this is made clear later in the paper (i.e., that they have slightly different coverage that you are trying to exploit). In this section, it might be nice to have a sentence listing the new data sets you are using, and mentioning why you are adding a second OMPS data set.

In the revised version, we indicate explicitly here updated and new datasets.

Page 4, line 9 – Fix parentheses on Boone reference

Corrected

Page 6, lines 14-16: You state the systematic uncertainty below 20 km and above 50 km here – what about 20 – 50 km?

We reformulated the sentence as follows: "The total systematic uncertainty is mainly related to cloud contamination and model errors in the lower stratosphere, and to the retrieval bias at high altitudes, with total values exceeding 5 % only above 50 km and below 20 km.

Page 8, eq 1: Is there a motivation for expressing anomalies as a fraction rather than absolute values (i.e., why divide by rho_m)? Can you briefly comment/justify this choice?

Ozone trends are usually presented in % decade$^{-1}$, therefore we selected such representation for visualization convenience. We added a short note in the revised version.

Page 8, lines 20-24: These two sentences are confusing. Is the second sentence referring to the new SAGE – CCI – OMPS+ data set, and saying you are doing something different than the original SAGE-CCI-OMPS data set (as described in the first sentence)? If so, you could start the sentence with something like "In SAGE-CCI-OMPS+, …"

Both sentences are about the original SAGE-CCI-OMPS dataset.

Page 9, line 6: "ACE-FTC" -> "ACE-FTS"

Corrected

Page 9, line 6 (and elsewhere): Why do you refer to version 4.1/4.2 of the ACE-FTS data? Shouldn't it be one or the other?

The version has this full notation: "4.1" and "4.2" are related to computational resources used for the processing ACE-FTS data.

Page 12, Figure 5: Panel a has a different x-axis, please make x-axes the same in this figure.

Corrected

Page 13, line 23: Drifts and steps are very different to me. Can you point more specifically to where possible steps occur.

Strong drifts or steps detected via comparisons with other data.  As observed in Figure S3, the difference between OMPS-UBr and MLS deseasonalized anomalies  in the tropics at 15 km changes abruptly  from positive to negative starting from 2018.

Fig. S3 caption: "NLS" -> "MLS"

Figs S5-S6: I think this is UBr OMPS. Please specify in captions.

Page 17, line 6: A colon is more appropriate here than a comma.

Page 18, line 3: "…only the data, which…" -? "… only data that …"

Corrected

Page 18, line 7: "As mentioned above…" – I don't see where it was mentioned that you exclude UBr data 14 – 16 km and 20S-20N. But maybe I'm missing something. Although you do identify UBr as problematic in this region, I think this is the first explicit mention that you are not using data in that region. Also, when I came to this sentence I began to question both the choice of masking region as well as the entire motivation for using UBr at all. It seems like the main motivation for using UBr is to get the additional data in the tropics that USask doesn't have, but then you say you are not actually using it here. Also, is the exact choice of masking really sufficient? It would be pretty easy to answer this by showing a latitude-altitude plot of the trend in the difference between UBr and USask. This would much more clearly help justify (or not) the choice of what levels and altitudes to remove from the analysis.

In the revised version, we removed the words "as mentioned above". Advantages of using both OMPS-LP datasets are (a) improved coverage (Figures 7 and 9; please note that only the 14-16 km region in the tropics is  excluded for OMPS UBr), and (b) better agreement with other datasets than if considered  separately (Figures 10 and S2).  The latitude-altitude plot of UBr and USask differences is shown already in Figure 7c. For detection of drifts, it is important to observe evolution of differences, which is illustrated in Figures 8, 9, 10, S2 and S3.

Page 23, line 17: What about the negative trends in the LS. It looks like some of those are significant, and that there are maybe a few more grid boxes that are significant in the new data set in comparison to the previous data set?

There are a few altitude-latitude bins where negative trends in the lower stratosphere are statistically significant. We added this note in the revised version.